# Diagnostic Process for Autism Spectrum Disorder: A Meta-Analysis of Worldwide Clinical Practice Guidelines for the Initial Somatic Assessment

**DOI:** 10.3390/children9121886

**Published:** 2022-12-01

**Authors:** Tom Dauchez, Guillaume Camelot, Charlotte Levy, Toky Rajerison, Kellen Briot, Adrien Pizano, Marie-Maude Geoffray, Loic Landrieu, Manuel Bouvard, Anouck Amestoy

**Affiliations:** 1Aquitaine Autism Ressources Centre, Centre Hospitalier Charles-Perrens, University Pole of Child and Adolescent Psychiatry, 33000 Bordeaux, France; 2Child and Adolescent Psychiatry Department, Centre Hospitalier le Vinatier, 69500 Bron, France; 3Health Services and Performance Research EA7425, Université Claude Bernard Lyon 1, 69100 Villeurbanne, France; 4LASTIG, Université Gustave Eiffel, IGN, ENSG, 94160 Saint-Mandé, France; 5Centre Hospitalier Charles Perrens, University Pole of Child and Adolescent Psychiatry, 33000 Bordeaux, France; 6Aquitaine Institute for Cognitive and Integrative Neuroscience (INCIA), UMR 5287, 33000 Bordeaux, France

**Keywords:** autism spectrum disorder (ASD), diagnosis, comorbidity, somatic assessment, guideline

## Abstract

(1) Background: Autism spectrum disorder (ASD) is a neurodevelopmental disorder that is highly associated with various somatic conditions that can be masked by the core symptoms of ASD and thus complicate the diagnosis. Identifying co-occurring somatic disorders is critical for providing effective healthcare and social services for ASD populations and influences their long-term outcomes. A systematic assessment of co-occurring somatic conditions is essential during this ASD diagnostic process. Therefore, this study aimed to identify the organization and content of the initial somatic assessment (ISA). (2) Methods: We conducted a systematic review of the clinical practice guidelines (CPG) for the ASD diagnostic process published between January 2005 and December 2019 in English and French and performed an appraisal following the Appraisal of Guidelines Research and Evaluation, second edition (AGREE-II). (3) Results: We selected 14 CPGs that were heterogeneous in quality, with methodological scores between 32.3 and 91.9. Clinical examinations are the first step in the ISA, and the participation of pediatric, neuropediatric, and genetic specialists was highly recommended by the majority of the CPGs. The recommendations included hearing screening tests (10/14), visual examinations (8/14), and systematic genetic investigations (4/14). The CPGs also described additional investigations that should be conducted based on numerous warning signs. (4) Conclusions: Screening for consensual international warning signs is necessary to perform a comprehensive and systematic ISA during the ASD diagnostic process. A “referral form” could be used to guide clinicians and improve the coordination process. This tool may reinforce epidemiological data on co-occurring somatic disorders in patients with ASD.

## 1. Introduction

Autism spectrum disorder (ASD) is an early onset and life-long neurodevelopmental condition characterized by difficulties with social interactions and nonverbal communication, as well as sensory abnormalities, stereotypic repetitive behavioral patterns, and fixated interests and activities, and may be associated with language and intellectual impairments according to the most recent Diagnostic and Statistical Manual (DSM-5) and International Classification of Diseases (ICD-11) [1,2]. ASD can be diagnosed in infants as early as 18 months and rarely develops after the age of 3 years. The etiology of autism is complex and involves the interplay of genetic and environmental factors. Meta-analyses of twin studies show that, despite high heritability estimates of 64–91%, shared environmental factors are also substantial (up to 35%), particularly when a narrow definition of autism is used [3,4,5].

The estimated global median prevalence of ASD is estimated to be between 0.62 and 0.70% but can reach 1% to 2% according to more recent large-scale surveys [3], with a large increase in prevalence over the last few years [5]. According to a new report from the Centers for Disease Control and Prevention, the incidence may be as high as 1 in 44 children, leading to an estimated prevalence of 2.3% [6] and a male-to-female ratio of approximately 3:1 [7]. Therefore, an ASD diagnosis is based on clinicians’ expertise and driven by structured and standardized tools and scales that are commonly and internationally validated and included in most international and European guidelines [8,9]. The term “spectrum” refers to the wide range of symptoms, skills, and levels of impairment that may characterize individuals with ASD.

While the ASD population is a heterogeneous group of individuals with various associated clinical features and prognoses, this heterogeneity is also known to be highly related to the presence of various co-occurring conditions. These conditions are both numerous and diverse, not only in terms of their nature (e.g., psychiatric, neurologic, genetic, metabolic, sleep-related, or eating disorders) but also in terms of their meaning, as some of these somatic disorders may be an etiology or a comorbidity of the ASD diagnosis as well as a differential diagnosis. In individuals with ASD, the reported rates of general medical, psychiatric, and neurodevelopmental co-occurring conditions can reach nearly 70% [3]. These conditions may include epilepsy (8–30%), gastrointestinal problems (9–70%), immune dysregulation (≤38%), genetic syndromes (~5%), and sleep disorders (50–80%) [3,10]. In parallel, Woolfenden et al. reported that individuals with ASD have a standardized mortality ratio (SMR) 2.8-fold that of individuals without ASD (95% CI [1.8–4.2]) [11], which is partly attributed to co-occurring somatic conditions [12,13]. Historically, individuals with ASD that had an initial somatic disorder that was considered a potential etiology of ASD symptoms (e.g., epilepsy) were qualified as “children with secondary autism” or “children with syndromic autism,” while individuals with ASD that had a co-occurring condition or abnormality that was not considered a possible etiology of the disorder (e.g., intellectual deficiency, ADHD) were categorized as having “complex autism.” In contrast, individuals with “isolated” or “pure” ASD had no somatic conditions at all. Since the publication of the most recent Diagnostic and Statistical Manual (DSM-5, 2013) and International Classification of Diseases (ICD-11, 2017) [1,2], ASD has been classified as a single nosological diagnosis that, depending on the co-occurring medical or genetic conditions, can have a better or worse prognosis.

Identifying co-occurring somatic disorders is critical for providing the most effective healthcare and social services to children and adults with ASD and influences long-term outcomes [11]. ASD significantly decreases a person’s educational, social, and employment opportunities and has been a public health priority in France since 2008 [14]. Children with autism are about twice as likely as their typical peers to experience chronic or recurrent pain due to somatic co-occurring conditions [15]. Various comorbidities of ASD, such as epilepsy [10,11], joint hypermobility-related disorders [16], gastrointestinal disorders [17], and sleep problems may be additional sources of pain [18]. Furthermore, individuals with ASD face significant inequities in healthcare [19,20] despite the high rate of medical comorbidities. As a result, these patients tend to have poorer health outcomes [21].

Exploration of somatic symptoms is recommended by numerous international guidelines as part of an early, standard ASD diagnostic process. However, among the various clinical practice guidelines (CPGs) available, the recommendations differ in terms of the systematic somatic investigations promoted. The aim of our study was thus to conduct a review of international guidelines in order to identify the most consensual recommendations in terms of the content of the initial somatic assessment (ISA) and disciplines involved.

## 2. Material and Methods

### 2.1. Literature Review

We performed a systematic review of the CPGs on the ASD diagnostic process published in English and French according to the definition of CPGs given by the Institute of Medicine: “statements that include recommendations intended to optimize patient care that are informed by a systematic review of evidence and an assessment of the benefits and harms of alternative care options” [22].

For the literature review, the Guideline International Network [23], MEDLINE and HTA databases (INAHTA), and web search engines (Google and Google Scholar) were searched for relevant literature published between January 2005 and December 2019. We extended the date range of the search to 2021 for CPGs relevant to the selection criteria. For these searches, the following keywords, MeSH terms, and text words were combined: “ASD” OR “autism”, OR “pervasive developmental disorder” AND “diagnosis” OR “diagnostic” OR “assessment” OR “evaluation”. We also conducted an identical search with the terms translating into French. The search was performed by two reviewers (D.T. and A.A.). The identified titles and abstracts were screened using the following inclusion criteria: a CPG on the ASD diagnostic process for children and adolescents aged <18 years published in English or French. The full texts of the documents that passed the screening stage were read to determine whether they met the study inclusion criteria. Further details are provided in Figure 1.

### 2.2. Quality Appraisal

The methodological quality of each selected document was assessed by three reviewers according to the methodological criteria of the Appraisal of Guidelines Research and Evaluation, second edition (AGREE-II) [24]. A unique reviewer (D.T.) rated all the CPGs. Additionally, two reviewers among a pool of 10 independent reviewers were selected to rate each document. Each participant reviewed at least three CPGs. The AGREE-II is an internationally recognized tool developed to address guideline quality variability. Four of the six domains of the AGREE-II (scope and purpose, stakeholder involvement, rigor of development, and editorial independence) were selected to assess the methodological rigor and transparency with which the guidelines were developed. Each of the 15 selected items was scored on a scale of 1 (strongly disagree) to 7 (strongly agree) (Table 1). The overall score is the sum of the four scaled domain scores (SDSs), which were determined for each domain using the following formula:SDS = (obtained score − minimum possible score)/(maximum possible score − minimum possible score)

Following the methodology proposed by Penner et al. [25], the reviewers held a meeting in January 2020 to address any instance in which the scores assigned by the reviewers were not within two points of agreement in order to reach a consensus. We measured the unbiased intraclass correlation coefficient (ICC) on all final scores.

### 2.3. Data Extraction

For this study, we performed a comparative analysis of the selected CPGs. For all CPGs, we specifically examined the co-occurring somatic disorders emphasized, the warning signs highlighted for clinicians, and the investigation processes by considering the content and disciplines involved in the ISA during the ASD diagnostic process.

## 3. Results

The search term screening returned 766 documents. Based on the abstracts, 24 documents were considered relevant to the definition of CPGs provided by the Institute of Medicine, and thus the full texts were reviewed. Finally, 14 CPGs that met the inclusion and exclusion criteria were included (Figure 2).

The scaled domain scores, overall scores, and rankings of all the included CPGs are presented in Table 2. The reviewers’ ratings showed strong agreement, with an intraclass correlation coefficient (ICC) of 0.89. Based on their overall scores, we characterized the CPGs into three groups (*excellent: score >* 80, *good: score >* 50, and *poor quality: score* < 50). In the first group, *excellent quality*, four CPGs with an overall score > 80 were included. The NICE Clinical Guidelines [9] and CRC National Guidelines [26] shared the same score of 91.9, the SIGN National Clinical Guidelines [27] had the third best score (90), and the HAS Good Practice Recommendations [8] had a score of 80.4. The full scores are listed in Table 2.

AGREE-II assessments of CPGs have been previously conducted by Penner et al. [25] and the CRC Administrative and Technical Report [28]. To evaluate the degree to which our findings were in agreement with these previous assessments, we calculated the ICC. An ICC of 0.79 was found between our assessment and Penner et al.’s assessment of the four CPGs included in that study (AAP 2007 [29], MIR [30], NICE, and AACAP [31]), and an ICC of 0.84 was found between our assessment and the CRC’s assessment, which included seven of the CPGs in our review (MIR, MISS [32], NICE, CONN [33], AACAP, NZ [34], and SIGN). The global agreement for the three appraisals included in both assessments was 0.65 (MIR, NICE, and AACAP). We also computed the ICC for each domain and observed the strongest agreement for the rigor of development domain (0.83 vs. Penner, 0.92 vs. CRC), and the weakest agreement for the scope and purpose (0.61 vs. Penner, 0.5 vs. CRC) and editorial independence (0.59 vs. Penner, 0.53 vs. CRC) domains.

### 3.1. Synthesis of Somatic Disorders Guidelines for Initial Somatic Assessment

Although the international CPGs were heterogeneous in quality, similar somatic disorders were documented, though they may have been categorized differently (etiological conditions, comorbidities, or differential diagnoses).

#### 3.1.1. Chromosome and Gene Abnormalities, Metabolic Diseases, and Mitochondrial Abnormalities

Numerous chromosome and gene abnormalities as well as metabolic diseases were reported in all CPGs (14/14), and unspecified mitochondrial abnormalities were cited in the majority of CPGs (9/14) (Table 3).

However, most of these conditions are very rare. To illustrate, fragile X syndrome (FXS) is the most common form of inherited learning difficulty and has been estimated to affect one in every 2500–7000 males and one in 2500–11,000 females [35]. Rett syndrome is now well known as a progressive neurological disorder that primarily affects girls, occurring in 1:10,000–15,000 live female births [36]. The worldwide prevalence is 1:10,000–1:23,000 female births [37]. The Sotos syndrome and the Smith Magenis Syndrom are also very rare. The birth incidences are estimated at 1:14,000 live births [38] and 1:25,000 births, respectively [39].

In a recent study, the prevalence of creatine transporter deficiency was 2.64% in individuals with neurodevelopmental disorders [40]. However, most of the metabolic disorders reported are also very rare. For example, the estimated incidence of guanidinoacetate methyltransferase (GAMT) deficiency, which is the second most common defect in the creatine metabolism pathway resulting in cerebral creatine deficiency syndrome (CCDS), in the general population ranges from 1:2,640,000 to 1:250,000 [41,42].

#### 3.1.2. Neurological Diseases

Higher rates of epilepsy and seizures were described in almost all the CPGs (13/14), except for the ACMG [34], which only included genetic evaluations. The two specific risk periods described for seizure and epilepsy in children and adolescents with ASD were preschool age and puberty [35], with an increased risk for females [36]. Epileptic encephalopathy, including Landau–Kleffner and West syndromes, was also cited by 10 of the 14 CPGs. The prevalence of epilepsy was greater in association with severity of intellectual impairment; 34% of children with autism and severe intellectual disability (IQ < 50) had epilepsy as compared to 27% of those with moderate ID (IQ 50–69) and 9% of those with IQ ≥ 70 [43]. In the UK, the prevalence was 2.80% in children aged 3–11 y and 4.10% in children aged 12–17 y [44]. The median overall period prevalence of epilepsy in people with autism was 12.1%, while the median overall period prevalence of autism in people with epilepsy was 9.0% when including all population types [10]. Over the course of childhood and early adolescence, the cumulative risk of developing epilepsy is approximately 6–7/1000 [45].

Numerous brain injuries, abnormalities, or conditions were cited by the CPGs, including degenerative central nervous system disorders (5/14), post-infectious cerebral diseases (e.g., measles encephalitis) (4/14), neonatal hypoxic-ischemic encephalopathy (4/14), cerebral palsy (4/14), traumatic brain injuries (3/14), and abnormal cerebral structures or cerebral malformations (2/14).

Macrocephaly was also reported to occur more frequently in the ASD population (5/14). It can affect 15 to 30% of children with ASD [29,46].

#### 3.1.3. Hearing Impairments and Visual Deficiencies

Hearing loss and hearing impairments were mentioned in all the CPGs, except for the ACMG [47] and CPS [48] (12/14).

Visual deficiency was only cited by 7 of the 14 CPGs. The prevalences of hearing impairment and visual deficiencies in ASD population are estimated to 4.90% and 14.9%, respectively in males [49].

#### 3.1.4. Sleep Disorders

Most of the CPGs (11/14) reported a frequent association between ASD and sleep disorders. Sleep disorders commonly include sleep onset insomnia, sleep maintenance insomnia, early awakening, parasomnias, sleep apnea, and restless legs syndrome.

Prevalence for sleep disorders varies from 2.08% in ASD females aged 0–18 y to 72.50% in ASD children with a mean age of 9 y [49]. The most prevalent subtypes of sleep disorders are sleep disorder–not otherwise specified (21.72%; *n* = 10,593), insomnia (9.52%), and sleep disorder breathing (9.26%) [50].

#### 3.1.5. Eating Disorders

Eating disorders were emphasized by most CPGs (10 of the 14 CPGs, four of which had an *excellent quality* rating [NICE, SIGN, HAS, CRC]).

The following two disorders were specifically highlighted: food selectivity, which is associated with an increased risk of nutritional deficiency (9/14) or obesity (2/14), and pica, which is associated with an increased risk of intoxication, including lead poisoning (5/14). The frequency of pica is reported to be higher in the ASD population (60%) than in the neurotypical 1–6-year-old population (10–32%) [51].

Atypical eating behaviors (e.g., limited food preferences and brand-specific preferences) occur much more often in autistic children (70.4%) compared to children with other disorders (13.1%) and children in the general population (4.8%) [52].

Binge eating disorders were not reported by any of the CPGs.

#### 3.1.6. Gastrointestinal Disorders

Gastrointestinal disorders were frequently reported by the CPGs (11/14). Numerous disorders were cited, among which constipation was the most common (7/14). The prevalence of gastrointestinal disorders in individuals with ASD remains unclear (9–70%) [53].

#### 3.1.7. Endocrine Diseases

Hypothyroidism or undefined endocrine disease was reported in four of the CPGs, two of which had an *excellent quality* rating (HAS and CRC).

#### 3.1.8. Auto-Immune Disorders

Two CPGs (NY [54] and AAP [55]) reported autoimmune disorders (allergies and eczema) as potentially associated with ASD, which may affect behaviors, especially feeding behaviors, and self-injury.

Allergies and autoimmune diseases are diagnosed significantly more often among children with autism than among children without autism (allergy: 20.6% vs. 17.7%; autoimmune disease: 1% vs. 0.76%), and asthma was diagnosed significantly less often (13.7% vs. 15.9%). Psoriasis occurred more than twice as often in cases than in controls (0.34% vs. 0.15%; OR = 2.35, 95% CI 1.36–4.08) [56].

#### 3.1.9. Toxins

Exposure to antenatal toxins was cited in most of the CPGs (11/14) as a factor associated with an increased prevalence of autism, as an etiologically associated factor, or as a differential diagnosis. Prenatal alcohol exposure was emphasized in six of the CPGs, antenatal exposure to valproic acid in four, exposure to thalidomide in three, and exposure to misoprostol in two of the CPGs.

Thalomide is a medication used nowadays to treat a number of cancers (including multiple myeloma), graft-versus-host disease, and a number of skin conditions including complications of leprosy. When first released, it was promoted for anxiety and trouble sleeping until it was first removed from the market in Europe in 1961, as it was responsible for many teratogenic deformities in children born. The birth defects caused by thalidomide led to the development of greater drug regulation and monitoring in most of the European and non-European countries.

Misoprostol is a prostaglandin analogue, used nowadays in medical abortion in the majority of countries, but it was also prescribed in the past and still in some countries, to prevent nonsteroidal anti-inflammatory drug-induced gastro-intestinal damages. It must not be taken during pregnancy as it has been shown to cause damage to the fetus or lead to miscarriage.

The association between ASD and antenatal exposure to selective serotonin reuptake inhibitors (SSRIs) was mentioned by two of the CPGs, but remains inconclusive.

#### 3.1.10. General Health Issues

Concomitantly, general health issues, including dental hygiene and care, which are comparable to those of the neurotypical population, were reported by most of the CPGs (8/14, two of which had an *excellent quality* rating [NICE and HAS]).

These general health issues can be complicated by the core symptoms of ASD, such as difficulties in communication, pain recognition, and expression. Underdiagnosed general health issues and pain can be a source of behavioral difficulties (including self-injury, feeding problems, or sleep disorders) and medical complications.

### 3.2. Synthesis of the Recommended Disciplines Involved

All CPGs included recommendations for the disciplines involved in the ASD diagnosis process except the ACMG, which only focused on the role of the medical geneticist (13/14). Most CPGs recommended that the ASD diagnosis process be performed by a multidisciplinary team (10/14), and only four of the CPGs recommend a graduated strategy in which a single clinician (e.g., sole experienced or trained pediatric care physician, experienced medical practitioner, selected psychologist) diagnoses the patient in simple cases (MISS, CONN, CRC, and CPS). However, all CPGs emphasized that systematic somatic evaluations should be conducted when ASD is suspected or diagnosed in a child (14/14). Eleven of the 14 CPGs (MIR, SIN [57], CONN, NICE, SIGN, NZ, NY, HAS, CRC, CPS, and AAP) recommended that the ISA be part of the initial diagnostic assessment and be conducted during the early diagnostic process, while the other CPGs recommended it be conducted “soon thereafter”.

A specific referral to a pediatrician (if not part of the initial multidisciplinary team) was recommended by five of the CPGs (including two of the four CPGs with *excellent quality* ratings [NICE and HAS]). The other CPGs indifferently assigned general practitioners or pediatricians to the initial somatic evaluation. Many additional specialists were included as potential referrals that could be involved in the initial somatic evaluation process (e.g., audiologists, ophthalmologists, pediatric neurologists, neurologists, geneticists, sleep physicians, gastroenterologists, dentists, and nutritionists).

We synthetized the consensus guidelines of the four high-quality CPGs (NICE, SIGN, HAS, CRC) in order to propose an international ISA meta-guideline for ASD (Figure 3).

#### 3.2.1. Medical Questioning and Investigations

Clinicians must collect an extensive amount of information from pre- and perinatal personal histories, including prenatal exposure to substances and toxins, birth records, newborn screening results, growth curve analyses, developmental and behavioral histories, learning or psychological difficulties, pharmacological treatment histories, and sleep and food analyses. Therefore, 11 of the CPGs (including the four with *excellent quality* ratings) recommended a complete exploration of the family medical history. Five of the CPGs (MISS, ACMG, CONN, AAP, and CPS) recommended that this be conducted over three generations.

#### 3.2.2. Physical Examination

All CPGs recommended performing a comprehensive physical examination including a growth analysis (height, weight, and cranial perimeter); general health screening; and neurological examination including oculomotor behaviors, signs of injury, morphological analyses, and skin examinations (14/14). Six of the CPGs recommended that Wood’s light be used for this purpose, including one with an *excellent quality* rating (NICE).

#### 3.2.3. Warning Signs

During medical questioning and physical examination, warning signs, including the family background, should be screened in children.

The most emphasized warning signs were neurologic abnormalities, including seizures (all CPGs), congenital or dysmorphic features (13/14, including the four with *excellent quality* ratings), regression in language or motor skills (13/14, including three of the CPGs with *excellent quality* ratings [NICE, SIGN, and HAS]), global developmental delay (GDD) or intellectual deficiency (ID) (11/14, including three with *excellent quality* ratings [NICE, SIGN and HAS]), skin abnormalities (10/14, including three with *excellent quality* ratings [NICE, SIGN and CRC]), head circumference abnormalities (10/14, including three with *excellent quality* ratings [NICE, SIGN and HAS]), lethargy or excessive fatigability (8/14, including one with an *excellent quality* rating [HAS]), cyclic vomiting (5/14), self-injury (3/14, including one with an *excellent quality* rating [NICE]), growth abnormalities (4/14), unexplained body organ dysfunction (3/14), oculomotricity abnormalities (3/14, including one with an *excellent quality* rating [HAS]), multiple miscarriages or perinatal deaths (1/14, with an *excellent quality* rating [HAS]), and premature ovarian deficiency (1/14). Many warning signs regarding sleep disorders were also reported by most of the CPGs. Further details are provided in Table 4.

#### 3.2.4. Hearing Tests and Visual Examinations

Hearing screening tests and complete hearing evaluations were recommended on a systematic basis by 10 of the CPGs, including two with *excellent quality* ratings (NICE and HAS). All other CPGs, except the ACMG, recommended that hearing tests be conducted in the case of any doubt in audition quality.

Visual examinations were recommended on a systematic basis by eight of the CPGs, including two with *excellent quality* ratings (NICE and HAS), while the other two CPGs with *excellent quality* ratings recommended that visual examinations be conducted at the slightest concern.

#### 3.2.5. Biological Testing including Metabolic Studies

Routine biological testing including metabolic studies were not recommended. All CPGs recommended conducting these investigations based on clinical indicators such as the warning signs described above. However, the ACMG recommended that the index of suspicion for metabolic disorders must remain high because of the associated “low incidence yet high impact”.

Five CPGs (MIR, MISS, SIN, CONN, and CPS) recommended conducting a blood lead level test in the presence of pica or a high-risk environment.

The AAP also recommended assessing for iron deficiency in the presence of pica, restless sleep, and night waking.

#### 3.2.6. Genetic Testing

Most of the CPGs did not recommend routine genetic testing, except for the CONN, NZ, and NY, which recommended that chromosomal microarray or karyotype (NZ) be performed. Furthermore, the ACMG, NZ, and NY recommend FMR1 testing for all boys without an etiology for ASD that systematically excluded fragile X syndrome.

Geneticist referrals for additional avide was widely recommended to target genetic testing (10/14 CPGs, including two with *excellent quality* ratings [SIGN and HAS]). In 2013, chromosomal microarray was preferred to karyotype in genetic evaluations by all CPGs.

#### 3.2.7. Electroencephalogram

Routine EEGs were not recommended by any of CPGs; however, EEGs were recommended by all the CPGs, except the ACMG, in the presence of clinical concerns. Two of the CPGs (one with an *excellent quality* rating [HAS]) also recommended a sleep EEG be performed in children.

#### 3.2.8. Magnetic Resonance Imaging

Similar to the EEG recommendations, routine MRIs were not recommended. Furthermore, the latest CPGs included a point of vigilance regarding the risk statement issued by the Food and Drug Administration in December 2016 regarding the extended or prolonged use of general anesthetic and sedative drugs in young children [58].

Two of the CPGs (one with an *excellent quality* rating [HAS]) recommended that the MRI be accompanied by a spectroscopy analysis when metabolic diseases are suspected.

## 4. Discussion

Although the clinical practice guidelines are heterogeneous in quality, they proposed relatively homogeneous content with regard to the ISA. The main disparities concerned recommendations related to the organization of the somatic assessment itself, the practices of routine genetic testing, and routine visual examination. In accordance with these results, multiple specialists are required to perform a systematic and comprehensive ISA during the diagnostic process. These specialists collect information through medical questioning and physical examination before deciding which further examinations are necessary, based on numerous consensual warning signs that should be screened for. In addition to the standardized scales already used in the diagnostic process for ASD assessments, additional tools are essential to address the specific somatic needs of each individual with ASD. This has important implications for educational settings as well as individual and parental concerns for improving quality of life and access to special services.

Following this review, our team is currently developing a French referral form, the ISA-ASD—Initial Somatic Assessment—to be used during the ASD diagnostic process (OBSI-TSA in French) to synthesize the steps for a systematic screening and referral to somatician colleagues in order to improve coordination between professionals. This is a coordination tool that could save time and limit the loss of information. The first version of the tool we developed (the OBSI-TSA in French) is being assessed by ongoing multicenter evaluations to analyze improvements in terms of screening for co-occurrences. We hope that these analyses will reinforce epidemiological knowledge on co-occurring somatic conditions in ASD among the French population.

Apart from these considerations and improvements, the high variability in the content of the ISA and disciplines involved reported by various European and non-European countries, as well as by clinicians within the same country, makes a group-based approach challenging and necessitates clear and consensual guidelines, general community communication, and clinician awareness for rapid appropriation and homogenization.

## Figures and Tables

**Figure 1 children-09-01886-f001:**
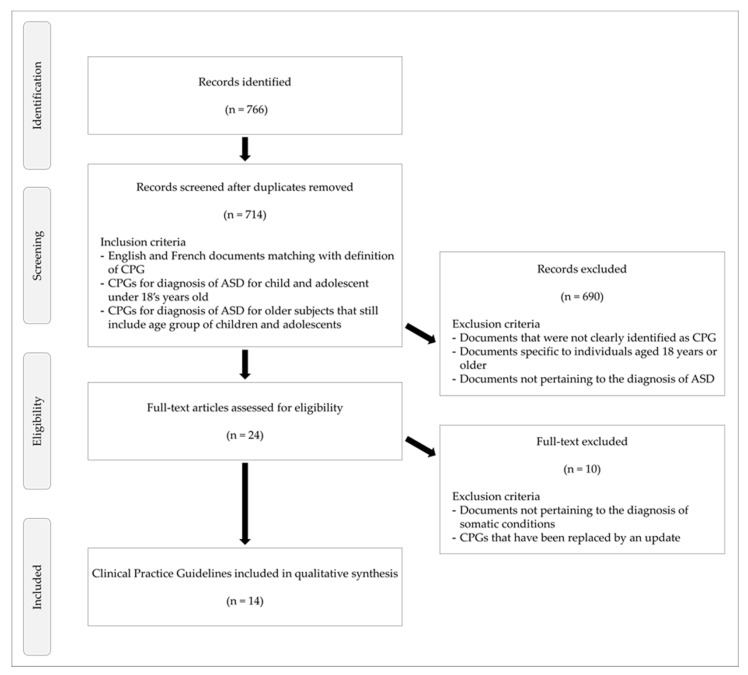
CPGs’ selection flow diagram.

**Figure 2 children-09-01886-f002:**
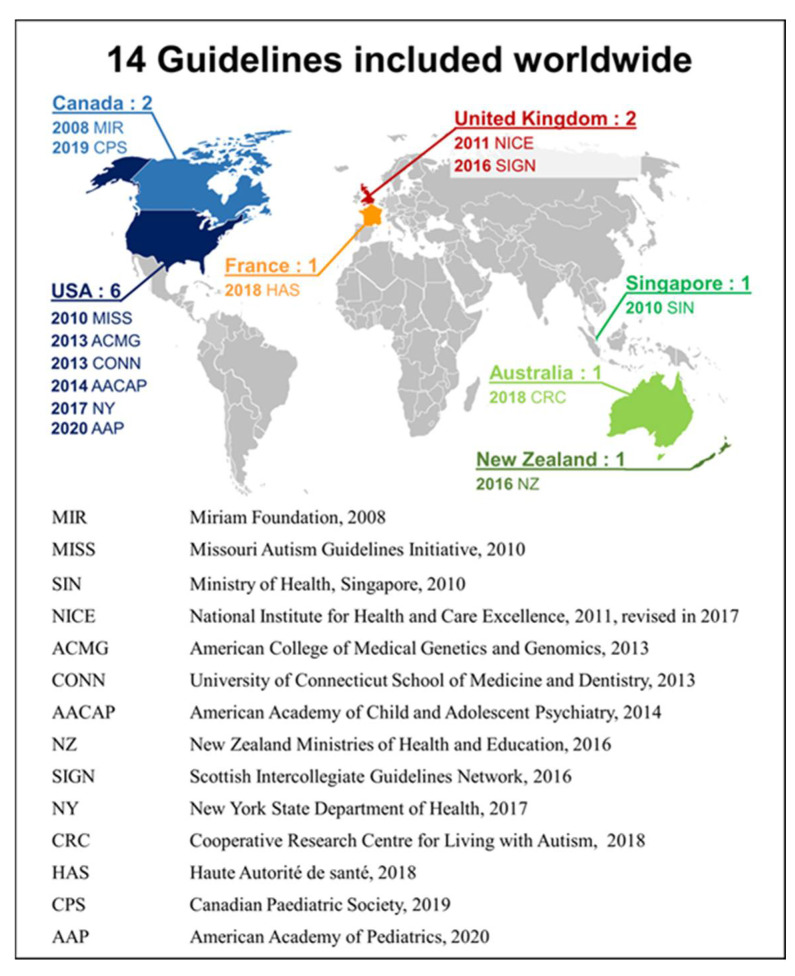
CPGs included and their country of origin.

**Figure 3 children-09-01886-f003:**
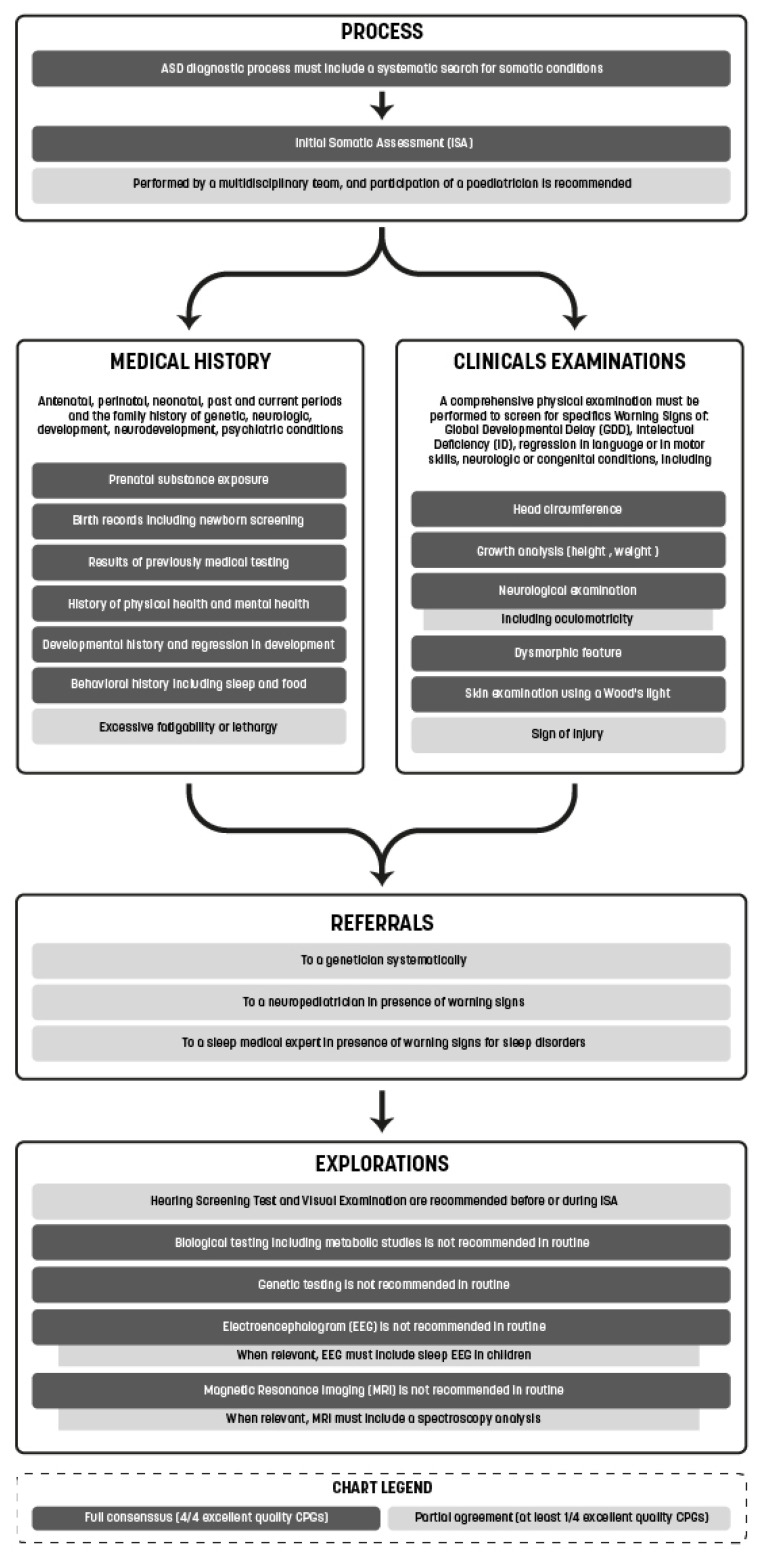
International meta-guidelines for initial somatic assessment (ISA) of ASD.

**Table 1 children-09-01886-t001:** The AGREE-II criteria used for quality appraisal.

AGREE II Criteria	Quality Appraisal Used Criteria
Domain 1. Scope and Purpose	Domain 1. Scope and Purpose: all criteria
Domain 2. Stakeholder involvement	Domain 2. Stakeholder involvement: all criteria
Domain 3. Rigour of development	Domain 3. Rigour of development: all criteria except for the “updating procedure” criterion
Domain 4. Clarity of presentation	Not evaluated
Domain 5. Applicability	Not evaluated
Domain 6. Editorial independence	Domain 4. Editorial independence: all criteria

**Table 2 children-09-01886-t002:** Quality appraisal: summary of AGREE-II domain and overall scores, and rankings for CPGs.

CPG	Domain 1		Domain 2		Domain 3		Domain 4		Overall	
	Score	Rank	Score	Rank	Score	Rank	Score	Rank	Score	Rank
MIR	88.9	5	83.3	4	52.4	7	27.8	12	62.6	6
SIN	85.2	6	57.4	9	30.2	9	13.9	14	44.4	9
MISS	83.3	8	63	8	22.2	10	30.6	11	43.7	10
NICE	100	1	92.6	3	87.3	2	94.4	1	91.9	1
ACMG	72.2	10	40.7	12	11.9	13	55.6	6	35.6	12
CONN	57.4	14	57.4	9	14.3	12	19.4	13	32.2	14
AACAP	72.2	10	33.3	14	58.7	5	50	8	55.2	8
NZ	85.2	6	81.5	5	51.6	8	55.6	6	64.8	5
SIGN	98.2	2	94.4	1	84.1	3	91.7	3	90	3
NY	74.1	9	72.2	7	54	6	47.2	10	60.7	7
CRC	94.4	4	94.4	1	88.9	1	94.4	1	91.9	1
HAS	96.3	3	79.6	6	74.6	4	77.8	4	80.4	4
CPS	66.7	13	33.3	14	7.1	14	66.7	5	33.2	13
AAP	72.2	10	44.4	11	21.4	11	50	8	40	11
*Excellent quality*				*Good quality*				*Poor quality*		

**Table 3 children-09-01886-t003:** Most often reported chromosome and gene abnormalities, and metabolic diseases.

Chromosome and Gene Abnormalities
Tuberous sclerosis complex (TSC1-TSC2): all CPGs
Rett syndrome (MECP2): all CPGs
Fragile X syndrome: all CPGs
Neurofibromatosis: 8/14
Down syndrome (trisomy 21): 7/14
Prader-Willi syndrome, Angelman syndrome and Dup15q syndrome (15q11-q13): 7/14
Phosphatase and tensin homolog (PTEN) gene mutation associated disorders (Cowden syndrome, Bannayan–Riley–Ruvalcaba syndrome): 7/14
Williams–Beuren syndrome (7q11.23): 5/14
Higher rates of Copy-number variations: 5/14
Phelan–MacDermid syndrome or 22q13.3 deletion syndrome (SHANK3): 4/14
Turner syndrome (45, X): 4/14
Cornelia de Lange syndrome (5p13.1-Xp11.22-p11.21): 4/14
Smith–Magenis syndrome (17p11.2): 4/14
Möbius syndrome (13q12.2-q13): 4/14
Muscular Dystrophy including Duchenne Muscular Dystrophy: 4/14
22q11.2 deletion syndrome (DiGeorge and Shprintzen syndromes): 4/14
CHARGE syndrome (8q12.1): 3/14
Sotos syndrome (5q35): 3/14
**Metabolic diseases**
Untreated Phenylketonuria: 7/14
Smith–Lemli–Opitz syndrome (3β-Hydroxycholesterol-7-reductase deficiency): 4/14
Disorders of creatine transport or metabolism: 4/14
Disorders of carnitine biosynthesis: 3/14
Mucopolysaccharidosis including mucopolysaccharidosis III: 3/14
Disorders of γ-aminobutyric acid metabolism: 2/14
Phosphoribosylpyrophosphate synthetase superactivity 2/14
Disorders associated with cerebral folate deficiency 2/14
Homocystinuria: 2/14

**Table 4 children-09-01886-t004:** Warning signs for co-occurring somatic conditions in the ASD population and the referrals and further examinations recommended.

**Warning Signs**	**Conditions That May Be Suspected**	**Referral, Further Examinations**
GDD and ID, dysmorphic features, congenital abnormalities	Chromosome and gene abnormalities	Referral for clinical genetics evaluation; could require oriented genetic tests
Microcephaly	Rett syndrome	
Male gender with drooling, recurrent respiratory infections, hypotonic facies		
Male gender	FXS	
Female gender with family history of X-linked compatible NDD, premature ovarian insufficiency, ataxia, or tremors		
Macrocephaly (>2.5 SD)	PTEN gene mutation, FXS	
Macrocephaly and excessive growth	Sotos syndrome	
Seizures or symptoms suggesting sub-clinical seizures	Epilepsy, epileptic encephalopathy	Pediatric neurologist or neurologist referral; could require EEG
Language regression	Epilepsy, epileptic encephalopathy including Landau-Kleffner syndrome	
Abnormalities on neurological examination including head circumference abnormalities	Neurological diseases	Pediatric neurologist or neurologist referral; could require EEG, MRI
Skin pigmentation abnormalities	TSC, NF1	Geneticist and pediatric neurologist or neurologist referral; could require genetic testing, EEG, MRI
Seizures, cyclic vomiting, lethargy, ataxia, hypotonia, dystonia, muscle weakness, dysmorphic features, skin abnormalities, poor growth, regression (particularly if associated fever, illnesses, or unusual odors), deafness, oculomotor abnormalities, metabolic acidosis, anemia, multi-organ system involvement, inadequate or absence of newborn screening	Metabolic and mitochondrial diseases	Pediatric neurologist or neurologist, specialists, geneticist referral; could require EEG, MRI, metabolic studies, genetic testing
Resistant sleep disorder, loud snoring, choking or periodic apnea during sleep	Obstructive sleep apnea, sleep disordered breathing	Pediatrician, otorhinolaryngologist, and sleep medicine services referral
Difficulties with sleep onset, night awakening, and parasomnias	Obstructive sleep apnea, restless legs syndrome, nocturnal seizures, medication side effects, circadian rhythm abnormalities	
Pica	Intoxications (e.g., lead poisoning)	Biological testing
Behavioural manifestations	General health issues, pain	Pediatrician or general practitioner referral

GDD: global developmental delay; ID: intellectual deficiency; NDD: neurodevelopmental disorders; FXS: Fragile X syndrome; SD: standard deviation; PTEN: Phosphatase and TENsin homolog; EEG: Electroencephalogram; MRI: Magnetic Resonance Imaging; TSC: Tuberous Sclerosis Complex; NF1: Neurofibromatosis type I.

## Data Availability

Not applicable.

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
