# Peer review of "Diagnostic Process for Autism Spectrum Disorder: A Meta-Analysis of Worldwide Clinical Practice Guidelines for the Initial Somatic Assessment"

_children, 2022, doi:10.3390/children9121886_

Round 1

Reviewer 1 Report

The findings also have important implicatons for educational settings and parents with ASD children.

Author Response

  • The findings also have important implications for educational settings and parents with ASD children.

We thank the reviewer and modified the discussion paragraph to add this consideration.

Please see the attachment : Revised Manuscript.

Reviewer 2 Report

Journal Review: “Initial Somatic Assessment in Autism Spectrum Disorder Diagnostic Process: A Meta-Analysis of Worldwide Clinical Practice”; submitted to Children.

A.    Overview

The manuscript consists of a systematic review of the more recent papers examining attentional strengths and weaknesses in several genetic disorders typically associated with intellectual disability. Using the framework of Ed Zigler’s developmental approach and the rigor required with regard to matching samples to truly understand attention deficits proved that the field is significantly behind best practices of conceptualizing and utilizing proper matching in order to detail profiles of attention across various domains. The paper provides a very well done look at the current literature, details attentional strengths and weaknesses in individuals with Williams syndrome, Down syndrome and Fragile X syndrome and provides important insight into how to better conduct studies in the future. Overall, there are very minor recommendations to strengthen the paper.

Comments:

Throughout the manuscript, if using DSM-5 as a reference, ASD should be autism spectrum disorder, not disorders.

Given that the authors first language is not likely English, it is recommended that the manuscript be reviewed for grammatical errors prior to resubmission.

Abstract: There are some grammatical errors in the abstract. For example., line 27 I believe the world ‘a’ is missing before population or population should be plural. Same with ‘during ASD diagnosis process’ should this be the diagnostic process? And diagnostic assessment? There are also tense changes throughout that should be reviewed for consistency. Also, ‘earing’ screening should be ‘hearing’ screening. There is also a ‘addressing questionnaire’ in the abstract. Not sure what this should be.

Introduction:

Page 2, line 38-54: The statement that ASD is recommended to be diagnosed before the 3rd birthday is not incorrect however I would recommend rephrasing that sentence to end it with and is only rarely developed after the age of 3. I would also split that sentence as it is very long.

Page 2, line 74: majoration should be majority

Page 2, line 81/82: why are some phrasings in quotation marks and others in chevrons?

Page 2, line 82 – what is meant by the term ‘consensual’

Line 182 – I believe ‘genic’ should be ‘genetic’

Results:

Each section of the co-occurring conditions should talk about prevalence as it is not that helpful to providers to simply know that each article is mentioning the condition. Age of children, if applicable, would also be helpful to know when to expect certain co-occurring conditions. As well, for some of the exposures, I.e., thalidomide, might be important to explain more given that this is not typically prescribed any more.  

In section 3.2 there is a discussion on the process of multidisciplinary team. It is not clear what is meant by a single clinician to conclude on the diagnosis of simple cases?

Check tables for words used, I.e., genic instead of genetic as well as grammar (clinicals examinations; to a genetician systematically)

Table 4 should be checked for word choice but it is a very helpful table

Author Response

  • Given that the authors first language is not likely English, it is recommended that the manuscript be reviewed for grammatical errors prior to resubmission.

We thank the reviewer and provided a professional review for grammatical errors, we hope it has improved the text.

  • Each section of the co-occurring conditions should talk about prevalence as it is not that helpful to providers to simply know that each article is mentioning the condition. Age of children, if applicable, would also be helpful to know when to expect certain co-occurring conditions

We thank the reviewer for this suggestion, and added the prevalence of the conditions and clarified the age of onset or the definitions if needed.

  • As well, for some of the exposures, I.e., thalidomide, might be important to explain more given that this is not typically prescribed any more.  

 We thank the reviewer for this suggestion, and added comment about reported treatments.

  • In section 3.2 there is a discussion on the process of multidisciplinary team. It is not clear what is meant by a single clinician to conclude on the diagnosis of simple cases?

 We thank the reviewer for this comment, and clarified the notion of single clinician

  • Check tables for words used, I.e., genic instead of genetic as well as grammar (clinicals examinations; to a genetician systematically)
  • Table 4 should be checked for word choice but it is a very helpful table

We thank the reviewer. The document has been professionally reviewed for language errors.

Please see the attachment : Revised manuscript

Reviewer 3 Report

This meta-analysis provides a useful summary using AGREE-II appraisal of clinical practise  guidelines for initial somatic assessment of patients with ASD.   The  ISAT-ASD tool discussed by the authors in the conclusion, once developed, will be very useful for screening patients to help identify co-occurring conditions. Although the AGREE-II tool provides an analysis of all components of the CPGs, could the authors make any comment on the suitability of the cut off date of 2005 used for the SLR - i.e could this date pull up outdated CPGs and recommendations in the field of ASD which is rapidly evolving? Although the NICE guidelines have been updated in 2017, some of the published CPG dates are from 2008 (MIR). 

Minor edits for authors:

Abstract:

1. Revise this sentence to remove the term 'key ingredient': 'Clinicals examinations remained the key ingredient to best complete the ISA'.

2. Also can the conclusion report on developing of the tool ISAT-ASD as a key outcome of the study?pti

Author Response

  • Although the AGREE-II tool provides an analysis of all components of the CPGs, could the authors make any comment on the suitability of the cut off date of 2005 used for the SLR - i.e could this date pull up outdated CPGs and recommendations in the field of ASD which is rapidly evolving? Although the NICE guidelines have been updated in 2017, some of the published CPG dates are from 2008 (MIR). 

We thank the reviewer for this comment. The first french CPG was proposed in 2005. We updated all CPG when applicable (the NICE for eg). Some of the CPG are outdated CPGs but the quality assessment of the CPGs took into account the age of the CPG and their updating processing.

                 - Revise this sentence to remove the term 'key ingredient': 'Clinicals examinations remained the key ingredient to best complete the ISA'.

We thank the reviewer, and rephrased the sentence.

                  - Also can the conclusion report on developing of the tool ISAT-ASD as a key outcome of the study?

We thank the reviewer for this comment,  we clarified this point in the discussion section, we hope it has improved the text.  

Please see the attachment : Revised manuscript

Round 2

Reviewer 2 Report

The authors took to heart the feedback from the reviewers and as a result this is a significantly stronger manuscript. 

Minor note: Line 297 - Thalidomide is spelled incorrectly